# A Deep Neural Network Model for Speaker Identification

**Feng Ye** [1,2] and **Jun Yang** [1,*]

1 Institute of Microelectronics of the Chinese Academy of Sciences, Beijing 100029, China; yefeng@ime.ac.cn
2 University of Chinese Academy of Sciences, Beijing 100049, China
* Correspondence: yangjun@ime.ac.cn

**Abstract:** Speaker identification is a classification task which aims to identify a subject from a given time-series sequential data. Since the speech signal is a continuous one-dimensional time series, most of the current research methods are based on convolutional neural network (CNN) or recurrent neural network (RNN). Indeed, these methods perform well in many tasks, but there is no attempt to combine these two network models to study the speaker identification task. Due to the spectrogram that a speech signal contains, the spatial features of voiceprint (which corresponds to the voice spectrum) and CNN are effective for spatial feature extraction (which corresponds to modeling spectral correlations in acoustic features). At the same time, the speech signal is in a time series, and deep RNN can better represent long utterances than shallow networks. Considering the advantage of gated recurrent unit (GRU) (compared with traditional RNN) in the segmentation of sequence data, we decide to use stacked GRU layers in our model for frame-level feature extraction. In this paper, we propose a deep neural network (DNN) model based on a two-dimensional convolutional neural network (2-D CNN) and gated recurrent unit (GRU) for speaker identification. In the network model design, the convolutional layer is used for voiceprint feature extraction and reduces dimensionality in both the time and frequency domains, allowing for faster GRU layer computation. In addition, the stacked GRU recurrent network layers can learn a speaker's acoustic features. During this research, we tried to use various neural network structures, including 2-D CNN, deep RNN, and deep LSTM. The above network models were evaluated on the Aishell-1 speech dataset. The experimental results showed that our proposed DNN model, which we call deep GRU, achieved a high recognition accuracy of 98.96%. At the same time, the results also demonstrate the effectiveness of the proposed deep GRU network model versus other models for speaker identification. Through further optimization, this method could be applied to other research similar to the study of speaker identification.

**Keywords:** speaker identification; speaker recognition; recurrent neural network; spectrogram; two-dimensional convolutional neural network; gated recurrent unit; mel-filterbank energy features; aishell-1

## 1. Introduction

Speaker recognition [1] is an important bio-feature recognition method. It is the task of recognizing the identity of someone based on the speaker's speech signal. Speaker recognition is a valuable biometric recognition technology and this method has been applied in several fields such as secure access to highly secure areas, machines such as voice dialing, banking, database, and computers. Due to the unique characteristics of speech signal, speaker recognition has drawn increasing attention from researchers in broad fields of information security for many years.

Speaker recognition study can be considered as the use of employing statistical methods to identify the individuals based on their unique acoustic properties, which are encoded in a sequence of successive samples in time. According to actual application, speaker recognition can be divided into two modes: Speaker verification and speaker identification [2–4]. The former study focuses on whether the claimed speaker is the true speaker, while the

latter aims at identifying the speaker. In this paper, we mainly conduct research on speaker identification.

Speaker identification is the task of determining an unknown speaker's identity by their speech signal, it is a 1:N match processing in which the extracted feature is compared against multiple templates [5]. In the speaker identification process [6], extracting speaker features from speech signals is a pivotal task. Human speech signals are a powerful medium of communication that contain rich information, such as gender, emotional traits, accent, etc. These unique characteristics enable researchers to identify speakers by voiceprint recognition [7]. The collected speaker utterances are fed into the deep learning network for training. In the recognition process, the speaker identification system matches the extracted speaker features with those in the model library. Then, the speaker with the highest probability of utterance is identified as the target speaker. However, the stability of speaker identification is not enough. The recognition depends on the length of the voice, the voice collection environment, and the physical condition of the speaker. In addition, the recognition performance is generally not satisfactory in a noisy environment.

Convolutional neural network (CNN) or recurrent neural network (RNN) models have performed well [8,9] in some speaker identification tasks, but we anticipate that the fusion of CNN and RNN models can benefit from different advantages due to their contrasting architectures. In this study, we propose a deep neural network (DNN) model based on CNN and gated recurrent unit (GRU) for the speaker identification. We compare its performance with some existing network models for the speaker identification task, including the Mel frequency cepstral cofficient-Gaussian mixture model(MFCC-GMM) model proposed by Maurya et al. [10], the CNN model proposed by Chen et al. [8], the RNN model proposed by Shahla et al. [9], and the multimodal Long Short-Term Memory (LSTM) model proposed by Jimmy [11]. In the process of proposing this model, we also tested other network models. The compared results among models are described in the results section of the paper. Moreover, our proposed network models for speaker identification were evaluated on the publicly available Aishell-1 datasets [12].

The main contribution of this paper is to propose a DNN model that combines the feature extraction advantages of CNN and the feature learning ability of GRU for time series information for speaker identification task. The reasons for CNN and GRU were chosen for model construction are as follows: 1. CNN is effective for spatial feature extraction and the spectrogram of speech contains spatial features of voiceprint, we chose CNN layers to extract features from the spectrogram. 2. Since deep networks can better represent long utterances than shallow networks and GRU has an excellent feature learning ability for time series information, and GRU has a better performance than traditional RNN, we chose stacked GRU layers in our model for frame-level feature extraction. Furthermore, it can be concluded that this method that combines the characteristics of CNN and RNN models could be effective on research similar to the study of speaker identification. Moreover, we compare its performance with existing network models for a speaker identification task and demonstrate the effectiveness of the proposed DNN model.

The rest of this paper is organized as follows. Related Works describes existing works on speaker identification. Section 2 presents the materials and methods of the proposed deep GRU network model, which mainly includes the following modules. A background overview of RNN and CNN are discussed in the background subsection. The proposed model is shown in the model overview subsection. The details of the dataset and data preprocessing are reported in the dataset and data preprocessing subsection. Section 3 presents the experimental results, with a description of the training process and evaluation metrics, and a comparison of several network models. Section 4 presents the conclusion of the study and describes the future work outlook.

*Related Work*

Research on speaker identification has a long history. In the 1940s, researchers began the study of speaker identification by conducting research on the spectrogram, but the effect

was unsatisfying. In 1956, several computer scientists put forward the concept of artificial intelligence [13], then speaker recognition began to enter the era of artificial intelligence research. At that time, researchers hoped to realize the identification of speakers through computer programs, but due to poor computer hardware capabilities and the immaturity of related algorithms, research on speaker identification did not achieve great results.

Until the 1980s, as a powerful branch in the field of artificial intelligence, machine learning [14] research began to use algorithms to analyze data, obtain relevant feature information from it, and then make decisions and predictions to solve the problem. Unlike traditional software programs designed to solve specific tasks, machine learning is trained with a large amount of data, using various algorithms to learn how to complete tasks from the data. The research on speaker identification has been greatly developed. The speaker identification process during this period usually consists of the following stages: Speech data preprocessing, feature extraction, speaker modeling, and scoring. Speech preprocessing includes pre-emphasis, framing, windowing, etc. Feature extraction is to extract the characteristic parameters that effectively characterize the speaker's traits from the speech signals. The typical features include mel-frequency cepstral coefficients [15], linear prediction coefficients [16], linear prediction cepstral coefficients [17], etc. Then speaker models are built through an analysis of these extracted features. At present, many methods are still used for speaker modeling such as gaussian mixture models (GMM) [5], GMM-universal background model [18], hidden Markov model [19], and the neural network model. The DNN model is discussed in this study.

In the past 10 years with the development of deep learning [20,21],CNN has been used in speaker identification tasks. In a neural network model, when given labeled data, a speaker recognition system based on deep learning extracts each speaker's deep features and performs supervised learning. Y. Lukic designed a convolutional neural network [22] which obtained an accuracy of 97.0% on the TIMIT dataset, corresponding to 19 misidentified speakers on a whole set of 630 speakers [22]. At the same time, LSTM networks [23] and traditional RNN [24] have been successfully applied to various sequence prediction and sequence labeling tasks. In language modeling, a traditional RNN model has obtained a significant reduction in perplexity over standard n-gram models [25]. By designing some gate structures, the LSTM network solves the long-term dependencies problem in traditional RNN and alleviates the vanishing gradient problem. Currently, the LSTM network has outperformed traditional RNN in language identification tasks [26]. LSTM models also perform better than traditional RNN models on learning context-free and context-sensitive languages [27]. For online and offline handwriting recognition, LSTM networks used together with a connectionist temporal classification (CTC) layer and trained from unsegmented sequence data, have been shown to outperform the hidden Markov model (HMM)-based system [28]. Similar techniques with a deep LSTM network have been proposed to perform grapheme-based speech recognition [29]. LSTM networks have also been proposed for phoneme prediction in a multi-stream framework for continuous conversational speech recognition [30]. However, the number of parameters for the LSTM network training is four times than traditional RNN, which is easy to cause overfitting. By further optimizing the structure, GRU has become a widely-used variant of RNN. The GRU network is similar to the LSTM network in performance, but the former network is faster to train and less likely to diverge. In current deep learning solutions, LSTM and GRU are often applied to tasks such as visual localization [31], traffic noise prediction [32], stock market forecast [33], and air pollution forecasting [34].

As mentioned above, since the speech signal is a continuous one-dimensional time series, most of the current research methods are based on CNN or RNN. Indeed, these networks perform well on some datasets [35], but there is no attempt to combine these two networks to study the speaker identification task. Due to the spectrogram that a speech signal contains, the spatial features of voiceprint and CNN are effective for spatial feature extraction and modeling spectral correlations in acoustic features [36], thus we chose CNN to extract features from the spectrogram. On the other hand, deep networks can better

represent long utterances than shallow networks [37], thus stacked GRU layers are used in the model for frame-level feature extraction, since they have been proven to be effective for speech processing [38,39]. The proposed model combining GRU and CNN should contain a better result.

## 2. Materials and Methods

### 2.1. Background

2.1.1. Recurrent Neural Network

In many recent studies, the RNN model has been a highly preferred method [40], especially for sequential data. A traditional RNN architecture is illustrated in Figure 1. Every node at a time step consists of an input from the previous node and it proceeds using a feedback loop. Each node produces a current hidden state and output by using the current input and previous hidden state as follows:

$$\mathrm{h}_t = f(U_h x_t + W_h h_{t-1} + b_h) \tag{1}$$

$$y_t = f(W_o h_t + b_o). \tag{2}$$

Here, $h_t$ denotes the hidden block of each time step $(t)$, $x_t$ is the input at some time step, $U$ and $W$ are the weights for the hidden layers in a recurrent connection, $b$ denotes the bias for hidden states and output states, and $f$ represents an activation function applied on each node throughout the network.

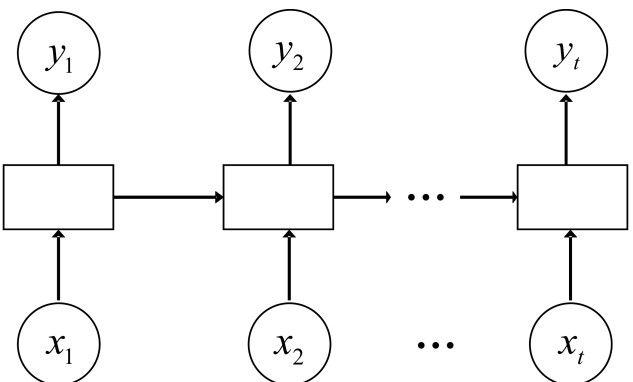

**Figure 1.** Conventional recurrent neural network (RNN) model architecture. The input is a time series, $x_t$ represents the input information at time $t$. The middle box represents the hidden state. $y_t$ represents the output at time $t$.

2.1.2. Long Short-Term Memory (LSTM)

The common drawback of a traditional RNN model is that, as the time step increases the network fails to derive context from the time steps of previous states much further behind. Such phenomenon is known as long-term dependency. Due to the deep layers of a network and recurrent behavior of a traditional RNN, exploding and vanishing gradient problems are also encountered quite often. Moreover, to address this problem, the LSTM model is introduced by deploying memory cells with several gates in a hidden layer [41]. Figure 2 illustrates the block of a hidden layer with an LSTM unit.

An LSTM network computes a mapping from an input sequence $x = (x_1, x_2, ...., x_T)$ to an output sequence $y = (y_1, y_2, .., y_T)$ by calculating the network unit activations using the following equations iteratively from $t = 1$ to $T$:

$$i_t = \sigma(W_{ix} x_t + W_{im} m_{t-1} + W_{ic} c_{t-1} + b_i) \tag{3}$$

$$f_t = \sigma(W_{fx} x_t + W_{fm} m_{t-1} + W_{fc} c_{t-1} + b_f) \tag{4}$$

$$c_t = f_t \odot c_{t-1} + i_t \odot g(W_{cx} x_t + W_{cm} m_{t-1} + b_c) \tag{5}$$

$$o_t = \sigma(W_{ox} x_t + W_{om} m_{t-1} + W_{oc} c_t + b_o) \tag{6}$$

$$m_t = o_t \odot h(c_t) \tag{7}$$

$$y_t = \phi(W_{ym}m_t + b_y). \tag{8}$$

Here, the $W$ terms denote weight matrices (e.g., $W_{ix}$ is the matrix of weight from the input gate to the input), $W_{ic}$, $W_{fc}$, $W_{oc}$ are diagonal weight matrices for peephole connections , the $b$ terms denote bias vectors ($b_i$ is the input gate bias vector), $\sigma$ is the logistic sigmoid function, and $i$, $f$, $o$, and $c$ are respectively the input gate, forget gate, output gate, and cell activation vectors, all of which are the same size as the cell output activation vector $m$, $\odot$ is the element-wise product of the vectors, $g$ and $h$ are the cell input and cell output activation functions, generally and in our experiment *tanh*, and $\phi$ is the network output activation function, Leaky ReLU in our experiments.

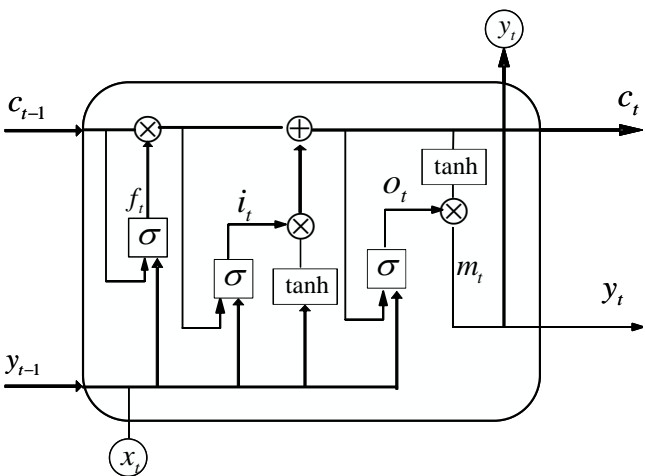

**Figure 2.** Long short-term memory (LSTM) architecture (a single memory block is shown for clarity).

2.1.3. Gated Recurrent Unit (GRU)

In the GRU cell unit, the two gates in the LSTM cell are combined into one gate [42]. In this architecture, the reset gate determines how to combine the new input with the previous stored information, and the update gate determines how much of the previous stored content works. In the absence of an output gate, it can be said that the GRU is a different implementation of the delivery and combination of the information. Compared with LSTM, the GRU network has made some simplifications in structure. A diagram of the GRU network structure is shown in Figure 3. The connection relationship in Figure 3 is given following equations:

$$r_t = \sigma(W_{xr}^T \cdot x_t + W_{yr}^T \cdot y_{t-1} + b_r) \tag{9}$$

$$z_t = \sigma(W_{xz}^T \cdot x_t + W_{yz}^T \cdot y_{t-1} + b_z) \tag{10}$$

$$\tilde{y}_t = \tanh(W_{x\tilde{y}}^T \cdot x_t + W_{y\tilde{y}}^T \cdot (r_t \otimes y_{t-1}) + b_{\tilde{y}}) \tag{11}$$

$$y_t = z_t \otimes y_{t-1} + (1 - z_t) \otimes \tilde{y}_t. \tag{12}$$

Here, $x$ is the input vector, $y$ is the output vector and $\tilde{y}$ is the candidate output vector, $W_{xr}$, $W_{xz}$, and $W_{x\tilde{y}}$ denote the weight matrices for the corresponding connected input vector, $W_{yr}$ , $W_{yz}$, and $W_{y\tilde{y}}$ represent the weight matrices of the previous time step, and $b_r$ , $b_z$, and $b_{\tilde{y}}$ are the bias. The GRU has two gates: $r$ is denoted as the reset gate and $z$ as the update gate.

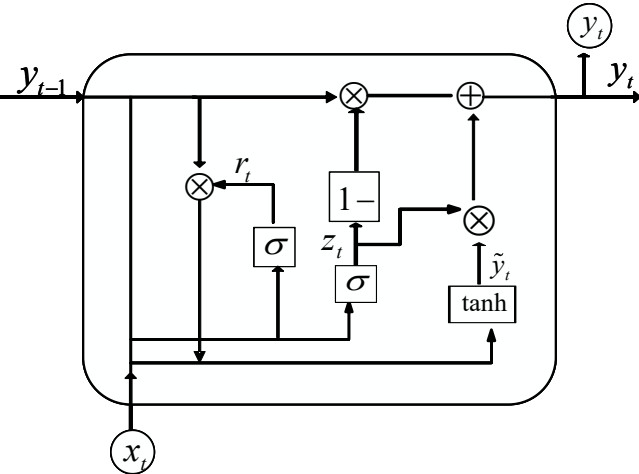

**Figure 3.** Gated recurrent unit (GRU) architecture (a single memory block is shown for clarity).

2.1.4. Convolutional Neural Network (CNN)

According to previous research, CNNs possess strong adaptability and gradually have become the main research tool in the field of image and speech [43,44]. In the study of speaker recognition, the spectrogram [45] gives a large amount of information including the personality characteristics of the speaker, and dynamically shows the characteristics of the signal spectrum change. Due to this feature of the spectrogram, it provides an effective method for researchers, thus feature vectors need to be obtained through the spectrogram. Although speech is a time-varying signal with complex correlations at a range of different timescales, the spectrogram provides a good solution. We use the spectrogram as the input of the CNNs, the spectrogram contains the identity information of the speaker and is a two-dimensional signal. At the same time, CNNs can provide translation invariance in time and space, so we can obtain the voiceprint features in the spectrogram space without destroying the time sequence. Therefore, this study proposes to use the spectrogram as the input of the convolutional neural network.

The convolutional layer of CNN contains multiple feature maps (Feature Map) [46]. The convolution kernel is essentially a weight matrix, the input is locally filtered through the convolution kernel. The CNN layer can effectively extract the structural feature from spectrogram and reduce the complexity of the model through weight sharing.

*2.2. Model Overview*

In this section, we mainly describe the proposed model in our study. The deep GRU network model combines the 2-D CNN and RNN based on the GRU cell unit. According to the characteristics of the spectrogram, in this model, the 2-D CNN layer is mainly used for feature extraction from the spectrogram. The RNNs-based GRU is for cyclic memory learning on the extracted features according to the relationship of voiceprint. The following content describes the entire architecture of the model in detail.

Deep GRU Architecture

In this study, we proposed a DNN model which combines 2-D CNN and GRU. The structure of our network model is summarized in Figure 4. In our proposed model, we first preprocess the speech to obtain the spectrogram. Then input the spectrogram into the convolutional layer and perform feature extraction. After the convolutional layer, GRU layers are stacked and times-series feature learning is performed in these GRU networks. Next, we obtain the unique voiceprint features of each speaker through the embedding operation, and finally the softmax layer [47] in the model will study and score the extracted features.

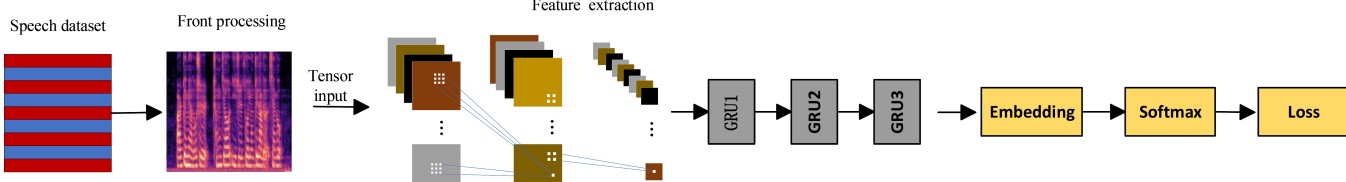

**Figure 4.** The structure of the deep GRU networks. The entire architecture diagram mainly includes voice signal preprocessing, two-dimensional convolutional layer, stacked GRU layers, a fully-connected layer, and softmax layer.

In our model, the input of the convolutional layer is a four-dimensional tensor which is a shape four-dimensional tensor in the form of $(batch\_size, filters, cols, channel)$, where $batch\_size$ is the number of samples selected for one epoch training, $filters$ represent the number of filters in the Mel filterbank, $cols$ represents the relevant features of each frame of speech, and $channel$ is the dimension of extracted features. Then, there is a $5 \times 5$ convolution kernel size, $2 \times 2$ stride of the convolution along the height and width. An average pooling layer is connected with the convolutional layer, its pooling size and stride are set as $2 \times 2$ . It is worth noting that we use batch normalization (BN) [48] between the convolutional layer and average pooling layer, and it accelerates model training. The BN transform is shown in Equations (13) and (14). In the algorithm, $\varepsilon$ is a constant added to the mini-batch variance for numerical stability:

$$\mu_\beta \leftarrow \frac{1}{m} \sum_{i=1}^{m} x_i; \sigma_\beta^2 \leftarrow \frac{1}{m} \sum_{i=1}^{m} (x_i - \mu_\beta)^2 \tag{13}$$

$$\widehat{x_i} \leftarrow \frac{x_i - \mu_\beta}{\sqrt{\sigma_\beta^2 + \varepsilon}}; y_i \leftarrow \gamma \widehat{x_i} + \beta \equiv BN_{\gamma,\beta}(x_i). \tag{14}$$

Here, $\mu_\beta$ is the mini-batch mean, $\sigma_\beta^2$ is the mini-batch variance, $\widehat{x_i}$ is the result of the normalization operation, $y_i$ is the result of scaling and shifting operations, and $\gamma$ and $\beta$ are parameters to be learned. Values of $x$ over a mini-batch: $\beta = \{x_{1...m}\}$ is the input, and $\{y_i = BN_{\gamma,\beta}(x_i)\}$ is the output.

Training DNN is complicated by the fact that the distribution of each layer's inputs changes during training, as the parameters of the previous layers change [48]. This slows down the backpropagation training, due to the nonlinearity caused by deep network and parameter saturation. BN normalizes the inputs and reduces nonlinearity of the system. As the result, BN stabilizes the distribution of activation values throughout training.

The recurrent network layer is built by stacking three GRU layers, their parameters settings for each layer are the same, and layer normalization (LN) is used between GRU layers. Unlike BN, LN [49] normalizes the input of all neurons in a certain layer of the deep network according to the following explanation.

In a traditional RNN, the summed inputs in the recurrent layer are computed from the current input $x^t$ and previous vector of hidden states $h^{t-1}$ which are computed as $a^t = W_{hh}h^{t-1} + W_{xh}x^t$ . The layer normalized recurrent layer re-centers and re-centers its activations using Formula (15):

$$h^t = f[\frac{g}{\sigma^t} \odot (a^t - \mu^t) + b]; \mu^t = \frac{1}{H} \sum_{i=1}^{H} a_i^t; \sigma^t = \sqrt{\frac{1}{H} \sum_{i=1}^{H} (a_i^t - \mu^t)^2}. \tag{15}$$

Here, $W_{hh}$ is the recurrent hidden to hidden weights and $W_{xh}$ are the bottom up input to hidden weights. $\odot$ is the element-wise multiplication between two vectors. $b$ and $g$ are defined as the bias and gain parameters of the same dimension as $h^t$.

In a standard recurrent network layer, there is a tendency for the magnitudes of the summed inputs of the recurrent units to either grow or shrink at every time-step [49]. This could lead to exploding or vanishing gradients. In a layer normalized RNN, the

normalization terms normalize the inputs to the layer, which results in much more stable hidden-to-hidden dynamics. In our network model, normalization is also used in stacked GRU layers and it can speed up network training.

Additionally, another difference in our experiment is that the Leaky Rectified Linear Unit (Leaky ReLU) [50,51] activation function was performed right after each layer before the output layer, many previous papers used Relu and tanh activation functions. At the same time, we chose to use categorical cross-entropy loss function in our study, and it can be computed by using the following formula:

$$Loss = -\frac{1}{N} \sum_{i=1}^{N} [y_i \ln a_i + (1 - y_i) \ln(1 - a_i)]. \tag{16}$$

Here, $N$ is the number of output size in the embedding layer, $y_i$ is the expected output vector of our model for $i$-th class, and $a_i$ is the actual output of the neuron. To obtain the output in a distributional manner across the entire subject, outputs $(y_1, y_2, ..., y_t)$ are computed by applying the non-linear softmax function to the output of the penultimate layer. The details of the proposed deep GRU network architecture are described in Figure 5.

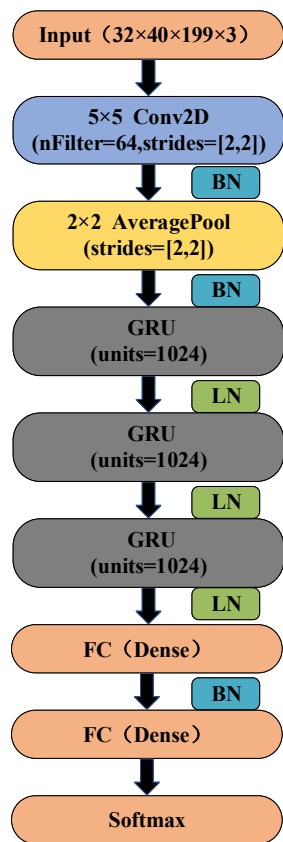

**Figure 5.** The architecture of the deep GRU network model, showing the input and some specific parameters of other layers, the framework is also the basis of code implementation.

In Figure 4, the input of the feature extraction network is formed by stacking information in each training frame with that in adjacent frames. 2-D CNN can extract spectrum distribution and differential spectrum distribution in a different frequency scale. GRU layers extract features of temporal spectrum distribution changes. Then, pooling and length normalization layers generate speaker embeddings. The high-dimensional statistics are converted into a single low-dimensional feature vector that encodes speaker identity.

Before we found the above proposed deep GRU network model, we also tried many different network architectures, including 2-D CNNs, a combination of CNN and LSTM cell unit network model, and a combination of the CNN and original RNN network model.

The experimental results of these models are unsatisfactory, and they will also be shown later for comparison.

*2.3. Experimental Setup*

2.3.1. Speech Dataset

To investigate the performance of the model in our study, we run speaker identification experiments with Aishell-1 dataset. Aishell-1 is a publicly available speech dataset, which consists of Mandarin speech recorded with a high-fidelity microphone (44.1 kHz, 16-bit). The audio recorded by the high fidelity microphone was down-sampled to 16 kHz and used to produce the Aishell-1 dataset. A total of 400 speakers from different accent regions in China participated in the recording. In our experimental setup, it was divided into the training set, validation set, and test set. The specific usage of the Aishell-1 dataset is shown in Table 1.

**Table 1.** The usage of the Aishell-1 dataset in experiments.

|  | **Speakers** | **Utterances** | **Ratio** |
|---|---|---|---|
| Training data | 400 | 127,551 | 90% |
| Validation data | 400 | 7278 | 5% |
| Test data | 400 | 7096 | 5% |

As a public speech data set, the Aishell-1 dataset is often used in related voiceprint recognition experiments. In this paper, we apply it to the speaker identification task. In the traditional way of machine learning, we often divide our data set by 60%, 20%, and 20%. Due to a large amount of data being available, we do not have to compromise for the data number. We used a greater portion to train the model [52]. In our experiments, we divided the data into a training set, validation set, and test set with the proportions of 90%, 5%, and 5%, respectively.

2.3.2. Data Preprocessing

In data preprocessing, our goal is to obtain the spectrogram, which contains rich acoustic features of speakers. The spectrogram is obtained by framing, windowing, short-time Fourier transform (STFT) [45], etc. The STFT is a process that computing the discrete Fourier transforms (DFT) of a signal over short overlapping windows. Short overlapping windows can maintain the short-time stability of speech signals in speech signal processing. The sampling frequency of the speech was 16 kHz, and the number of FFT points in this experiment was 512. The discrete STFT of the framed speech signal is calculated by the following equation:

$$X[k] = \sum_{n=0}^{N-1} x[n]e^{-j\frac{2\pi nk}{N}} \quad k = 0, 1, 2, ...., N-1. \tag{17}$$

Here, $x[n]$ denotes the framed signal with $n$ as the sequence number in the frame, $N$ denotes the frame size. The power spectrum is defined by using Formula (18):

$$P(k) = |X(k)|^2. \tag{18}$$

In our experiment, after loading an audio file, a floating-point time series was obtained. After its energy spectrum was calculated, then it was converted onto the Mel scale [53]. Additionally, more analysis can be extended after converting a power spectrum (amplitude squared) to decibel units. In this process, the frame length was set to 32 milliseconds, the window shift between successive frames is 16 milliseconds, and the window function selects the Hamming window. Finally, we got the spectrogram as shown in Figure 6, and the specific process of preprocessing is shown in Figure 7.

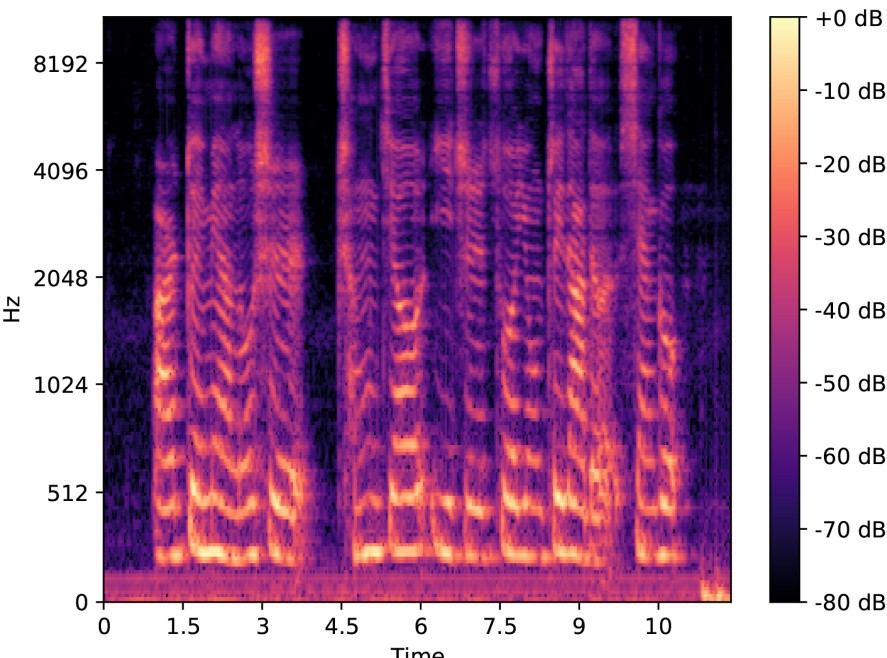

**Figure 6.** Spectrogram of speech. The intensity of any given frequency component at a given time is expressed by color depth. Darker, smaller; brighter, larger.

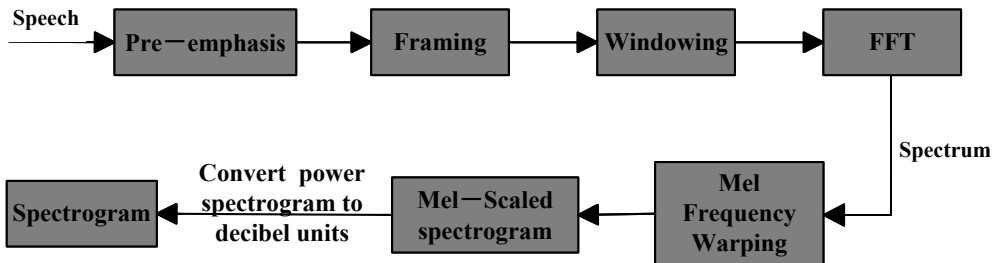

**Figure 7.** The process of the data preprocessing. The purpose of the preprocessing process is to obtain a spectrogram of the speech signal for further analysis.

The spectrogram combines the advantages of the time domain and frequency domain. At the same time, it obviously contains the change of speech spectrum along with time. As shown in Figure 6, the x-axis of the spectrogram is time, the y-axis is frequency, and the brightness represents amplitude: Darker, smaller; brighter, larger. The energy distribution of different frequency signals can be seen in the spectrogram. Additionally, the intensity of any given frequency component at a given time is expressed by color depth. Different degrees of black and white on the spectrogram form different lines, which are called voiceprint [7].

Finally, we compute delta features from the vector sequence, which is to calculate the feature vectors based on preceding and following N frames, the result is called the first-order features and the second-order features when N is 1 and 2.

In our study, the first-order features and second-order features are combined with the log Mel-filterbank energy features into a numpy [54] array. After performing some mathematical processing on feature vectors, a multidimensional tensor is sent into the convolutional layer for feature extraction. Then, the model is trained through iterative computations.

## 3. Results

### 3.1. Training

In order to speed up the training procedure, our proposed network schemes are implemented using TensorFlow deep learning library written in Python, which can be executed on a graphics processing unit (GPU). Due to parallel processing, GPU is generally more efficient than a central processing unit (CPU) on neural network processing. Our experiments are conducted on a dedicated Nvidia TITAN Xp GPU. In 2-D CNN and RNN-LSTM network models training stage, we choose the gradient descent algorithm with a linear decreasing learning rate schedule from 0.0001 to 0.00001. The model is trained for 80 epochs with a minibatch size of 32. Training pairs are re-shuffled in each training epoch. The validation set is used for hyper-parameter tuning and early stopping.

In the training phase for the deep GRU network model, the batch size is determined as 32, and the optimization is deployed by the Nadam optimizer [55,56], which incorporates Nesterov momentum into Adam. The initial learning rate is set at 0.0001. The loss function is selected as the categorical cross-entropy as mentioned in (16), where $y_i$ is the truth target vector, and $a_i$ is the output vector of the model for $i$-th class. The learning rate and convolutional kernel size of the network are conducted with various settings by trial-and-error approach, and the optimal setting for each network that yields the best performance results is chosen. The weights in the models were initialized randomly at the start of the training process, and progressively updated throughout the process. At the same time, to address the difficult problem of overfitting in deep learning model training, the dropout layer is also applied in the 2-D CNN layer and the GRU layer. In this study, a dropout of 0.3 at the 2-D CNN layer of the networks, and a dropout of 0.2 at the stacked GRU layers were deployed to avoid an overfitting problem typically encountered in deep neural networks. Moreover, layer normalization in GRU layers and batch normalization in convolutional layers and dense layers were deployed to avoid an overfitting problem typically encountered in deep neural networks. An example of the deep GRU network model training progress is given in Figure 8.

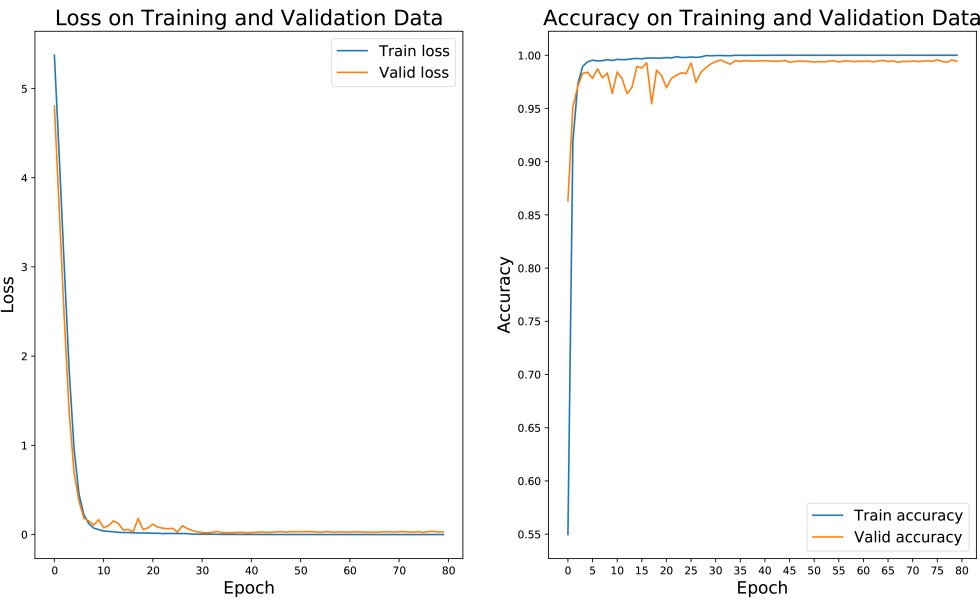

**Figure 8.** Loss and accuracy of the deep GRU network model with the Aishell-1 dataset. The left block presents loss vs. training epochs and the right block represents the accuracy vs. training epochs.

In order to verify the feature extraction ability of the CNNs proposed in our experiments, we specially designed a 2-D CNN network model for speaker identification experiments. In the following picture, the model training results are shown in the following picture. It can be seen from Figure 9 that the 2-D CNN network model used in the

experiment has converged after approximately 66 rounds of training and the results of the validation set are also relatively ideal. One training progress of the 2-D CNN network model is given in Figure 9.

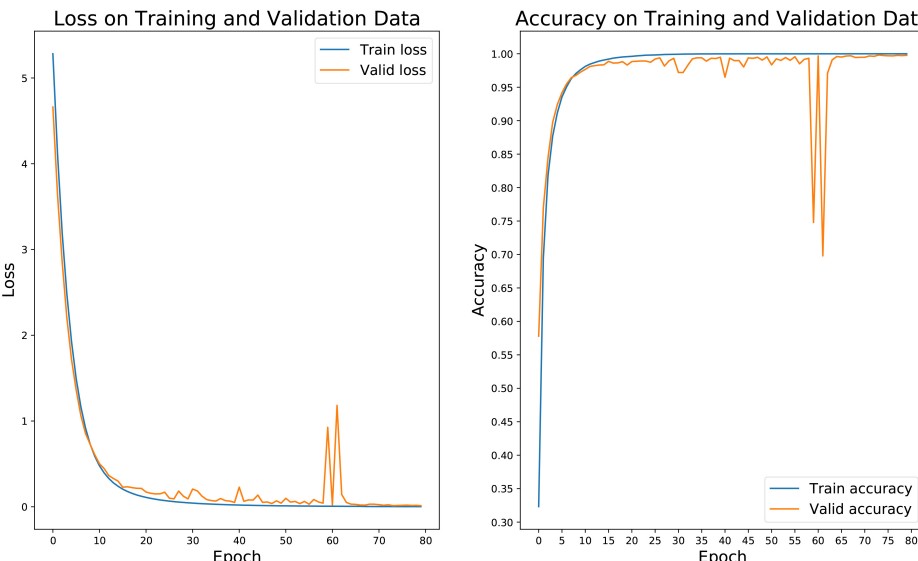

**Figure 9.** Loss and accuracy of the 2-D convolutional neural network (CNN) network model with the Aishell-1 dataset. The left part of the figure describes the loss vs. training epochs, and the right part describes the accuracy vs. training epochs.

### 3.2. Evaluation Metrics

The performance evaluation of speaker verification system mainly depends on two parameters, namely false acceptancy rate (FAR) and false rejection rate (FRR) [4]. FAR refers to an error in identifying a non-target speaker as a target speaker. FRR is an error caused by misrecognizing the target speaker for a non-target speaker. However, in a closed set speaker identification system, the correct recognition rate (accuracy rate) and top-N correctness [57] are commonly used to evaluate the performance of the system. Recognition rate refers to the probability of that the speech to be recognized can be correctly found the corresponding speaker from the target set. The speech to be recognized is usually identified as the speaker with the greatest similarity to the target speaker set, and its recognition accuracy ratio can also be called the top-1 recognition accuracy rate. There is another evaluation method called top-N recognition accuracy rate: If the N recognition speakers with the largest similarity include the correct speaker. Then the recognition result is considered to be correct.

Since in this speaker identification task, the accuracy rate refers to the probability of correctly identifying the speaker. The total number of successfully recognized voices (TNSV) is divided by the total number of tested voices (TNTV), and it can be formulated as below:

$$Accuracy = \frac{TNSV}{TNTV}. \tag{19}$$

### 3.3. Model Results

For the experiments, the total of 141,925 utterances from the Aishell-1 dataset was divided into a 90% training set, 5% validation set, and 5% test set. To visualize the learning efficiency of the models, one training progress of the proposed deep GRU network model with the Aishell-1 dataset is shown in Figure 8, where the left block presents loss vs. training epochs and the right part represents the accuracy vs. training epochs. To evaluate our proposed deep GRU model, we also conducted experiments with a 2-D CNN network

model, traditional RNN, and LSTM network models, and mainly discussed the experiment results using the Aishell-1 dataset and Gaussian white noise-added Aishell-1 dataset.

Table 2 reports the overall accuracy of different models we tried in our different research periods. The reported overall accuracy ranged from 95.42% to 98.96% with the Aishell-1 dataset, the proposed deep GRU model achieved the highest recognition accuracy in all network models. Figure 10 shows the above results in a column diagram format, it can be seen from the comparison that the proposed deep GRU network model has high robustness. We also conducted some experiments with several existing successful network models for comparison, and the experiment results are shown in Table 3. It is clear that when Gaussian white noise was added into the testing data, the recognition accuracy of other models dropped significantly; contrastingy, the proposed deep GRU network model still achieved a robustly high recognition accuracy.

**Table 2.** Performance of classification accuracy of models. The noise-added Aishell-1 in Table 2 is the same as the Aishell-1 with Gaussian white noise added in Table 3, and it refers to the clean Aishell-1 dataset with Gaussian white noise added.

| Types of Model | Datasets | Overall Accuracy (%) | Real Time per Epoch |
|---|---|---|---|
| 2-D CNN | Aishell-1 | 95.42% | 1035 s |
| | noise-added Aishell-1 | 48.57% | |
| CNN + original RNN | Aishell-1 | 96.67% | 1154 s |
| | noise-added Aishell-1 | 55.29% | |
| CNN + RNN-LSTM cell | Aishell-1 | 97.09% | 1065 s |
| | noise-added Aishell-1 | 70.42% | |
| Proposed Deep GRU | Aishell-1 | 98.96% | 980 s |
| | noise-added Aishell-1 | 91.56% | |

**Table 3.** Performance comparison with several existing models.

| Methods | Datasets | Overall Accuracy (%) |
|---|---|---|
| Proposed Deep GRU | Aishell-1 | 98.96% |
| | Aishell-1 with Gaussian white noise added | 91.56% |
| MFCC-GMM [10] | Aishell-1 | 86.29% |
| | Aishell-1 with Gaussian white noise added | 55.29% |
| CNN [8] | Aishell-1 | 92.45% |
| | Aishell-1 with Gaussian white noise added | 62.39% |
| RNN [9] | Aishell-1 | 89.67% |
| | Aishell-1 with Gaussian white noise added | 58.29% |
| Multimodal LSTM [11] | Aishell-1 | 96.25% |
| | Aishell-1 with Gaussian white noise added | 70.42% |

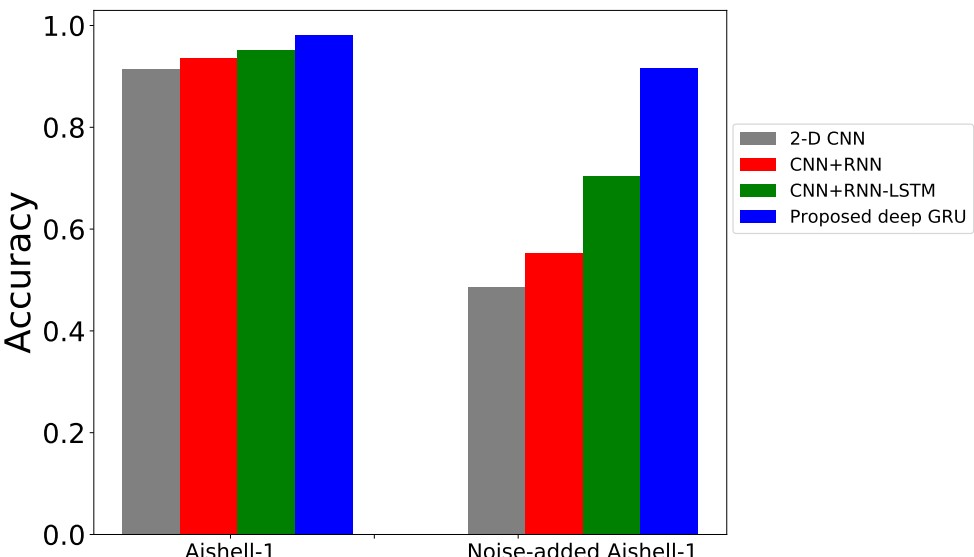

**Figure 10.** Performance comparison of proposed deep GRU network model with other network models. The block on the left side shows the performance of the models on the original Aishell-1 dataset, and the right part corresponds to the performance on the Aishell-1 dataset with Gaussian white noise.

During our experiments, we tried to use RNN and its variants in the model. After the results, we made further analysis on the parameters of the model and training time. Tables 4–6 list the detailed structure information and number of training parameters for those models we tried in our different research periods. Table 2 shows the time consumptions of one epoch training for different models. In these models, the trainable parameters of the CNN + original RNN model are the least, and the CNN + LSTM model has 50% more parameters than the deep GRU model. Due to structural improvements, we can clearly see that the deep GRU model's average training time of each epoch is the least, while training the CNN + original RNN model is the most time-consuming. This point proves that our deep GRU model has great advantages in structure.

**Table 4.** The details on deep CNN + traditional RNN network model.

| Layer Name | Struct | Stride | Param# |
|---|---|---|---|
| Input | $32 \times 40 \times 199 \times 3$ | - | - |
| 2-D Conv | $5 \times 5$ kernel, 64 filter | $2 \times 2$ | 4.8 k |
| Average pooling | $2 \times 2$ pooling | $2 \times 2$ | 0 |
| RNN | 1024 cell | - | 2.4 m |
| RNN | 1024 cell | - | 2.1 m |
| RNN | 1024 cell | - | 2.1 m |
| average | - | - | 0 |
| Dense | $1024 \times 512$ | - | 0.52 m |
| Length Normalization | - | - | 0.2 m |
| Output (softmax) | $1024 \times 400$ | - | 0 |
| Total | - | - | 7.32 m |

**Table 5.** The details on deep CNN + RNN-LSTM unit network model.

| Layer Name | Struct | Stride | Param# |
|---|---|---|---|
| Input | $32 \times 40 \times 199 \times 3$ | - | - |
| 2-D Conv | $5 \times 5$ kernel, 64 filter | $2 \times 2$ | 4.8 k |
| Average pooling | $2 \times 2$ pooling | $2 \times 2$ | 0 |
| LSTM | 1024 units | - | 9.4 m |
| LSTM | 1024 units | - | 8.3 m |
| LSTM | 1024 units | - | 8.3 m |
| average | - | - | 0 |
| Dense | $1024 \times 512$ | - | 0.52 m |
| Length normalization | - | - | 0.2 m |
| Output (softmax) | $1024 \times 400$ | - | 0 |
| Total | - | - | 26.93 m |

**Table 6.** The details on the deep GRU network model.

| Layer Name | Struct | Stride | Param# |
|---|---|---|---|
| Input | $32 \times 40 \times 199 \times 3$ | - | - |
| 2-D Conv | $5 \times 5$ kernel, 64 filter | $2 \times 2$ | 4.8 k |
| Average pooling | $2 \times 2$ pooling | $2 \times 2$ | 0 |
| GRU | 1024 units | - | 5.1 m |
| GRU | 1024 units | - | 6.3 m |
| GRU | 1024 units | - | 6.3 m |
| average | - | - | 0 |
| Dense | $1024 \times 512$ | - | 0.52 m |
| Length normalization | - | - | 0.2 m |
| Output (softmax) | $1024 \times 400$ | - | 0 |
| Total | - | - | 18.42 m |

## 4. Conclusions

In this article, we briefly introduced the research on the speaker identification task. Considering 2-D CNN has its advantage on feature extraction of a 2-D structure and voice spectrogram contains rich voiceprint information, we proposed a deep RNN model based on GRU combining a 2-D CNN layer for speaker identification. The proposed models integrate both advantages of feature extraction of 2-D CNN and temporal dependency of the GRU cell unit. In deep RNN networks based on the GRU cell unit, we further conducted cyclic memory learning on time series in the network to obtain hierarchical distinct features that represent each speaker in the embedding process. At the last layer, the softmax classifier layer in the network was used to learn and score the features to obtain the results of speaker identification.

In comparison to the performance of these network models on the Aishell-1 speech dataset, the proposed deep GRU network model achieved the best performance in terms of model training time, recognition accuracy, and stability. However, the disadvantage of the proposed method is the time cost in the training phase, and the stability of the model under complex noise environments is worth further study. Thus, our future work will aim to improve the robustness of the model in general practical environments.

**Author Contributions:** Conceptualization, F.Y.; Formal analysis, F.Y.; Methodology, F.Y.; Project administration, F.Y.; Software, F.Y.; Validation, J.Y.; Writing—original draft, F.Y.; Writing—review & editing, J.Y. All authors discussed the results and contributed to the final version of the manuscript. All authors have read and agreed to the published version of the manuscript.

**Funding:** This research received no external funding.

**Institutional Review Board Statement:** Not applicable.

**Informed Consent Statement:** Not applicable.

**Data Availability Statement:** The speech dataset used in this paper is an open source Mandarin speech corpus, and is 178 h long. It is a part of AISHELL-ASR0009, of which the utterance contains 11 domains, including smart home, autonomous driving, and industrial production. The whole recording was put in a quiet indoor environment, using three different devices at the same time: A high fidelity microphone (44.1 kHz, 16-bit,); Android-system mobile phone (16 kHz, 16-bit); and iOS-system mobile phone (16 kHz, 16-bit). Audios in high fidelity were re-sampled to 16 kHz to build AISHELL-ASR0009-OS1. A total of 400 speakers from different accent areas in China were invited to participate to the recording. The manual transcription accuracy rate is above 95%, through professional speech annotation and strict quality inspection. The corpus is divided into training, development, and testing sets. (This database is free for academic research, not in the commerce, if without permission.) This dataset is publicly available at http://www.aishelltech.com/kysjcp, accessed on 29 May 2019.

**Conflicts of Interest:** The authors declare no conflict of interest.

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
