# Peer review of "A Deep Neural Network Model for Speaker Identification"

_applsci, doi:10.3390/app11083603_

Round 1
Reviewer 1 Report
Speaker identification is a classification task that aims to identify a subject from a given time-series sequential data. In this paper, we propose a deep recurrent neural network (RNN) model based on the gated recurrent unit (GRU) combining a two-dimensional convolutional neural network (2-D CNN) for speaker identification. Structurally, GRU cell in RNN deploys an update gate and a reset gate in a hidden state layer. Due to the reduction of gates, this structural change is computationally efficient than a conventional long short-term Memory (LSTM) network. In addition, a convolutional layer is added into the proposed network model for local feature extraction, so that the recurrent network layer with GRU cell can learn key voiceprint features faster. The paper is interesting overall, but following are the comments that must be addressed:
Comments:
- A major contribution of the paper looks very weak authors need to explain as in the current version on page 2 Lin 53~54 it looks very confusing for readers.
- The authors need to draw the block diagram of the proposed approach step by step.
Aishell-1 speech dataset usage will be imbalanced how to tackle imbalanced data to train deep learning algorithm. The authors did not mention it in the paper???
- Figure 4 Feature extraction part authors need to explain which features extracted and how any specific features or all features??
- The authors need to explain why he used A Deep GRU Network Model for Speaker Identification Based on Recurrent Neural Networks in contrast with other deep learning algorithms?
- Authors missing experiment setup??
- Why is Layer Norm Basic LSTM Cell much slower and less accurate than LSTM Cell? As the authors mentioned Unlike BN, Layer Normalization (LN) normalizes the input of all neurons in a certain layer of the deep network. Line 196 page 6, need to explain it more.
- Related work and introduction section need to update with updated CAD, AI, ML,DL papers such as.
Gelly, G., Gauvain, J. L., Le, V. B., & Messaoudi, A. (2016). A Divide-and-Conquer Approach for Language Identification Based on Recurrent Neural Networks. In INTERSPEECH (pp. 3231-3235). Nagrani, A., Chung, J. S., & Zisserman, A. (2017). Voxceleb: a large-scale speaker identification dataset. arXiv preprint arXiv:1706.08612. Shafik, A., Sedik, A., Abd El-Rahiem, B., El-Rabaie, E.S.M., El Banby, G.M., Abd El-Samie, F.E., Khalaf, A.A., Song, O.Y. and Iliyasu, A.M., 2021. Speaker identification based on Radon transform and CNNs in the presence of different types of interference for Robotic Applications. Applied Acoustics, 177, p.107665. M. A. Khan and Y. Kim, "Deep learning-based hybrid intelligent intrusion detection system," Computers, Materials &
Continua, vol. 68, no.1, pp. 671–687, 2021. https://www.techscience.com/cmc/v68n1/41825Shafik, Amira, Ahmed Sedik, Basma Abd El-Rahiem, El-Sayed M. El-Rabaie, Ghada M. El Banby, Fathi E. Abd El-Samie, Ashraf AM Khalaf, Oh-Young Song, and Abdullah M. Iliyasu. "Speaker identification based on Radon transform and CNNs in the presence of different types of interference for Robotic Applications." Applied Acoustics 177 (2021): 107665.
There are few formatting issues please adjust it especially align all Figures with text according to MDPI format.
Author Response
Dear sir,
As for your suggestions, I used the "track changes " in the Microsoft Word to give a ponit-by-point reply, please check the attachment. Thank you for your comments,
Sincerely,

Reviewer 2 Report
This paper proposed a recurrent neural network model called GRU combining a two-dimensional convolutional neural network (2-D CNN) in order to the speaker identification. The experimental outcomes revealed that the proposed model performed a high recognition accuracy of 98.96% with Aishell-1 dataset. The paper is well-written, well-organised and the accuracy results are meaningful. However, the paper requires some major modifications and after applying them, it can be published in this journal. The recommendations are as follows:
1. The abstract should be reduced and the general information of RNNs would be better to remove. And also please briefly discuss the main challenges of the speaker identification problem.
2. In the introduction, please list the main contributions of this paper and the research question, and motivations.
3. In the related work section, initially please answer this research question, what are the benefits and drawbacks of LSTM and GRU? add some relevant references about the successful application of LSTM and GRU in other scientific fields such as:
a) Hybrid Neuro-Evolutionary Method for Predicting Wind Turbine Power Output. arXiv preprint arXiv:2004.12794, 2020.
b) Bi-LSTM model to increase accuracy in text classification: combining Word2vec CNN and attention mechanism. Applied Sciences, 10(17), 5841. 2020.
C) Air pollution forecasting based on attention‐based LSTM neural network and ensemble learning. Expert Systems, 37(3), e12511.2020.
4. In figure 8 and 9, the loss function convergence rate is high and looks like a pre-mature convergence, please test the smaller values of the learning rate at 10^-6 until 10^-9 and plot both loss and accuracy training manners.
Author Response
Dear sir,
As for your suggestions, I used the "track changes " in the Microsoft Word to give a ponit-by-point reply, please check the attachment. Thank you for your comments,
Sincerely,
Feng Ye

Round 2
Reviewer 1 Report
The authors resolve my previous queries this paper still needs a lot of improvement.
Authors need to re-write the Abstract in a more meaningful way example (Problem definition=> How existing methods are lacking => proposed solution => Outcome
related work section structure is very poor authors should redraw the structure as AI>ML>DL> then proposed approach with AI, should include these papers
Gelly, G., Gauvain, J. L., Le, V. B., & Messaoudi, A. (2016). A Divide-and-Conquer Approach for Language Identification Based on Recurrent Neural Networks. In INTERSPEECH (pp. 3231-3235). Nagrani, A., Chung, J. S., & Zisserman, A. (2017). Voxceleb: a large-scale speaker identification dataset. arXiv preprint arXiv:1706.08612. Shafik, A., Sedik, A., Abd El-Rahiem, B., El-Rabaie, E.S.M., El Banby, G.M., Abd El-Samie, F.E., Khalaf, A.A., Song, O.Y. and Iliyasu, A.M., 2021. Speaker identification based on Radon transform and CNNs in the presence of different types of interference for Robotic Applications. Applied Acoustics, 177, p.107665. M. A. Khan and Y. Kim, "Deep learning-based hybrid intelligent intrusion detection system," Computers, Materials &
Continua, vol. 68, no.1, pp. 671–687, 2021. https://www.techscience.com/cmc/v68n1/41825Shafik, Amira, Ahmed Sedik, Basma Abd El-Rahiem, El-Sayed M. El-Rabaie, Ghada M. El Banby, Fathi E. Abd El-Samie, Ashraf AM Khalaf, Oh-Young Song, and Abdullah M. Iliyasu. "Speaker identification based on Radon transform and CNNs in the presence of different types of interference for Robotic Applications." Applied Acoustics 177 (2021): 107665
in the introduction section at Line57 authors discuss the major contribution which looks very weak major contribution, authors should think about it deeply and elaborate it more.
There are many formating and space in the paper authors need to fix it.
The authors need to explain why he used A Deep GRU Network Model for Speaker Identification Based on Recurrent Neural Networks in contrast with other deep learning algorithms> AUTHORS should explain it more technically.
Table 1. Details of Aishell-1 dataset, authors select training, testing and validation 320,20,40 respectively i=any complexity issue??
Table 2. Performance of classification accuracy of Models should be merged with Table 7. Models training time.
Author Response
Dear Reviewer,
Thank you very much for your comments, and I revised it point-by-point based on your suggestions.
Please see the attachment.
Sincerely,
Feng Ye

Reviewer 2 Report
The authors did not highlight the new modifications. Each applied modification must be highlighted to recongnise.
Author Response
Dear reviewer,
Thank you very much for your comments.
Regarding for your comments, I used the "Track Changes" functions in Mircrosoft Word, and each applied modification was highlighted to recongnise.
Please check the revised content in the attachment (you are the reviewer 2 in word), thank you!
Wishing you a beautiful day.
Kind regards,
Mr Feng Ye

Round 3
Reviewer 1 Report
The authors did excellent work resolve all my previous comments. Now this paper looks very good and interesting for the readers so I agree to accept this paper in present form.
Reviewer 2 Report
The paper can be published after modifying the English.